# A Synthetic Peptide, CK2.3, Inhibits RANKL-Induced Osteoclastogenesis through BMPRIa and ERK Signaling Pathway

**DOI:** 10.3390/jdb8030012

**Published:** 2020-07-09

**Authors:** John Nguyen, Semaj Kelly, Ryan Wood, Brian Heubel, Anja Nohe

**Affiliations:** Department of Biological Sciences, University of Delaware, Newark, DE 19716, USA; njohn@udel.edu (J.N.); shkelly@udel.edu (S.K.); ryanwood@udel.edu (R.W.); brianph@udel.edu (B.H.)

**Keywords:** osteoporosis, casein kinase 2, BMP2, CK2.3, osteoclastogenesis, BMPRIa

## Abstract

The skeletal system plays an important role in the development and maturation process. Through the bone remodeling process, 10% of the skeletal system is renewed every year. Osteoblasts and osteoclasts are two major bone cells that are involved in the development of the skeletal system, and their activity is kept in balance. An imbalance between their activities can lead to diseases such as osteoporosis that are characterized by significant bone loss due to the overactivity of bone-resorbing osteoclasts. Our laboratory has developed a novel peptide, CK2.3, which works as both an anabolic and anti-resorptive agent to induce bone formation and prevent bone loss. We previously reported that CK2.3 mediated mineralization and osteoblast development through the SMAD, ERK, and AKT signaling pathways. In this study, we demonstrated the mechanism by which CK2.3 inhibits osteoclast development. We showed that the inhibition of MEK by the U0126 inhibitor rescued the osteoclast development of RAW264.7 induced by RANKL in a co-culture system with CK2.3. We observed that CK2.3 induced ERK activation and BMPRIa expression on Day 1 after stimulation with CK2.3. While CK2.3 was previously reported to induce the SMAD signaling pathway in osteoblast development, we did not observe any changes in SMAD activation in osteoclast development with CK2.3 stimulation. Understanding the mechanism by which CK2.3 inhibits osteoclast development will allow CK2.3 to be developed as a new treatment for osteoporosis.

## 1. Introduction

The skeletal system plays an important role in the development and maturation processes of vertebrates. It plays roles in many biological processes such as movement, support, blood cell production, calcium storage, and endocrine regulation [1]. The development of the skeletal system involves two major types of bone cells: osteoblasts and osteoclasts. Osteoblasts are cuboidal-shape-like cells that develop from mesenchymal stem cells (MSC). The development of osteoblasts is regulated by several signaling pathways including the bone morphogenetic protein (BMP) signaling pathway [2]. The development of osteoblasts is regulated by transcription factors such as runt-related transcription factors 2 (RUNX2), distal-less homeobox 5 (Dlx5), and osterix (OSX) [3,4]. Osteoclasts are developed from hematopoietic stem cells (HSCs) [5,6]. Under the stimulation of stem cell factor (SCF), interleukin (IL)-3, and IL-6, HSCs give rise to common myeloid progenitors (CMPs). Then, granulocyte/macrophage colony stimulating factor (GM-CSF) promotes the differentiation of CMPs into granulocyte/macrophage progenitors (GMPs). Macrophage colony stimulating factor (M-CSF) further promotes GMP differentiation into the monocyte/macrophage lineage, which is considered to be osteoclast precursors [6]. The development of osteoclast precursors is supported by osteoblasts. Two important differentiation factors for osteoclasts, M-CSF and receptor activator of nuclear factor kappa-β ligand (RANKL), are secreted by osteoblasts [7,8,9,10,11,12,13]. The receptor for M-CSF on osteoclast precursors is the colony-stimulating factor-1 receptor (c-fms) [14]. Activated c-fms induces the activation of the tyrosine kinase domain, which transphosphorylates specific tyrosine residues on the cytoplasmic tail to increase its affinity for SH2 domain-containing signaling residues [6,15]. For example, phosphorylated Y559 and Y721 interact with proto-oncogene tyrosine-protein kinase Src (c-Src) and phosphoinositide 3-kinase (PI3K), respectively, resulting in the activation of the AKT signaling pathway [6,16,17,18]. By contrast, the phosphorylation of Y697 and Y974 recruits Grb2 and subsequently activates the ERK signaling pathway [6,19]. The receptor for RANKL is receptor activator of nuclear factor kappa-β (RANK). Activated RANK recruits TNF receptor associated factor 6 (TRAF6) to form a signaling complex containing c-Src, TAK1, TAB1, and TAB2, which subsequently activate the AKT, NF-κB, and MAPK signaling pathways [12,20,21,22].

The skeletal system is maintained and repaired by the bone remodeling process. Every year, about 10% of the skeletal system is replaced by the process [23]. The balance between osteoclast activity and osteoblast activity is highly regulated during bone remodeling. This balance is kept until the peak of skeletal development at around 25 years of age. Due to the aging process, the balance is shifted toward bone resorption more than bone formation [24]. As a result, the skeletal system becomes deteriorated, and bone mass declines with age. Osteoporosis (OP) is characterized by a significant loss of bone mineral density (BMD) that will eventually result in bone fractures. There are no symptoms for osteoporosis. Thus, OP often progresses without being noticed until patients experience the first fracture. The mortality rates after a hip fracture within a year are 30% and 32.7% for women and men, respectively [25]. The estimated treatment cost for osteoporotic fractures by 2025 is 27 billion USD [26].

Bone morphogenetic protein 2 (BMP2) has been identified as an important factor in bone formation and skeletal development. Studies on BMP2 show that BMP2 is essential for the process of bone repair and fracture healing [27,28,29,30]. BMP2 signaling is mediated through BMP receptor type 1a (BMPRIa) and BMP receptor type 2 (BMPRII) [31,32,33]. The osteogenic ability of BMP2 has been shown to be mediated by both SMAD and non-SMAD signaling pathways [34]. BMP2 was approved by the FDA to treat spinal fusion and open tibia fractures [35]. However, treatment with BMP2 is not without risks [36,37,38]. Our research focuses on fine tuning the BMP2 signaling transduction to improve its potential to treat bone fractures. Our research first identified casein kinase 2 (CK2) as an activator of BMP2 signaling transduction [39,40]. We designed a novel peptide CK2.3 to mimic the bone morphogenetic protein receptor type Ia (BMPRIa) binding site (amino acids 213–217) for CK2, allowing it to bind to CK2 to block the interaction between BMPRIa and CK2 [39]. CK2.3 was shown to promote bone formation and suppress osteoclast development in vivo [40,41,42]. It was suggested that CK2.3 induces mineralization in myoblast C2C12 cells though the activation of ERK [43]. Similarly, an elevated activation of ERK was observed in the femurs of 8-week-old mice injected with CK2.3 [41]. The underlying mechanism of CK2.3’s suppression of osteoclast development is still unknown.

As mentioned before, MAPK and NF-κB are shown to play important roles in osteoclast development that induced by RANKL [12,20,21,22]. The importance of p38 MAPK in osteoclast development is demonstrated by the inhibition of RANKL-induced osteoclast development in RAW264.7 cells by dominant-negative forms of p38 and MKK6 [44]. The activation of ERK has been suggested to play an important role in osteoclast development [45,46,47,48,49]. Erk1 knockdown in bone marrow mononuclear cells has been shown to decrease the formation of osteoclasts [46]. Additionally, the activation of p38 MAPK and ERK/MAPK has been demonstrated to be downstream of both RANKL and BMP2 in osteoclastogenesis [44,45,46]. The role of NF-κB in the inhibition of osteoclast development has been shown by the double knockout of p50 and p52, two important components for NF-κB activation [50]. NF-κB activation in osteoclast development is followed by the interaction of RANKL with RANK [51]. Additionally, crosstalk between BMP2 signaling and NF-κB signaling has been shown through the demonstration of the interaction between the p65 subunit of NF-κB and Smad4 [52]. Combining these findings suggests that p38 MAPK, ERK MAPK, and NF-κB play an important role in osteoclast differentiation mediated by osteoclast differentiation factors such as RANKL and BMP2 (Figure 1).

In this study, we elucidated the mechanism by which CK2.3 inhibits osteoclast development. We showed that CK2.3 downregulated osteoclast gene markers such as TRAP, DC-STAMP, and Cstk in the blood sera of 6-month-old retired breeder mice. We showed that the effect of CK2.3 inhibiting the osteoclast development of RAW264.7 cells could be nullified by the MEK inhibitor U0126, suggesting that CK2.3 acted through the MEK/ERK signaling pathway. This was further confirmed by immunofluorescence and Western blotting, which showed that CK2.3 upregulated ERK activation in the femurs of 6-month-old retired breeder mice and in RAW264.7 cells, respectively. We also showed that CK2.3 at 2.3 µg/kg, the concentration that we showed to be the most beneficial, was stable in the femurs for up to 4 weeks [42]. These results were the first to demonstrate the mechanism by which CK2.3 inhibits osteoclast development and could further advance the development of CK2.3 as a new treatment for osteoporosis.

## 2. Materials and Methods

### 2.1. Animals and Ethical Approvement

Retired breeder female C57BL/6 (B6) mice that were 6 months old were obtained from the Jackson Laboratory (Bar Harbor, ME). The mice were injected in the tail vein once a day for five consecutive days with CK2.3 (GeneScript) at 2.3 µg/kg or 50 µL of phosphate-buffered saline (PBS). Wild-type B6 mice, 6 months old, were a gift from Dr. Lachke’s laboratory at the University of Delaware. This study was approved by the Institutional Animal Care and Use Committee (University of Delaware, Newark, DE, USA) and has the following approval number: AUP#1194.

### 2.2. Cell Culture

Monocyte/macrophage RAW264.7 cells were obtained from the American Type Culture Collection (Manassas, VA, USA). The RAW264.7 cells were plated on a 6-well plate at 1.8 × 10^4^ cell/cm^2^ in Dulbecco’s Modified Eagle Medium (DMEM) (Krackeler Scientific, Cat# 23-90-013-PB) supplemented with 1.8 g/L of sodium bicarbonate, 2 mM l-Glutamine (Gemini Bio-Products (West Sacramento, CA, USA), Cat# 400-106), 1 mM sodium pyruvate (Cellgro (Herndon, VA, USA), Cat# MT-25-000-Cl), and 100 U/L of penicillin and 0.1 mg/L of streptomycin (Gemini Bio-Products, Cat# 400-109). After plating, the cultured plate was kept in the incubator at 37 °C with 5% CO_2_ for 24 h. Then, the osteoclast differentiation factor RANKL at 10 ng/mL (Sino Biological (Beijing, China), Cat# 11682-HNCH) and different concentrations of CK2.3 (50 nM, 100 nM, 500 nM, or 1000 nM) were added into the culture medium (Day 0). The culture medium with fresh RANKL and CK2.3 added was changed on Day 3. Stimulation was terminated on Day 5. The cells were fixed with 4% paraformaldehyde for 20 min at room temperature and stained for tartrate resistant acid phosphatase (TRAP) using an Acid Phosphatase Leukocyte (TRAP) kit (Sigma-Aldrich (St. Louis, MO, USA), Cat# 387A-1KT). Osteoclasts were stained purple when observed under a brightfield microscope and counted in each treatment.

### 2.3. Inhibitor Treatment

RAW264.7 cells were plated on a 6-well plate at 1.8 × 10^4^ cell/cm^2^ in DMEM supplemented with 1.8 g/L of sodium bicarbonate, 2 mM l-Glutamine, 1 mM sodium pyruvate, and 100 U/L of penicillin and 0.1 mg/L of streptomycin. After plating, the cultured plate was kept in the incubator at 37 °C with 5% CO_2_ for 24 h. Then, the RAW264.7 cells were serum starved. After 18–20 h of starvation, the cells were treated with the MEK inhibitor U0126-EtOH (U0126) at 1 µM, NF-κB inhibitor caffeic acid phenethyl ester (CAPE) at 1 µM, or p38 inhibitor SB202190 at 10 µM or left untreated. After 2 h of inhibitor pre-treatment, the medium was changed. The cells were treated as follows: RANKL, RANKL + inhibitor (U0126, CAPE, or SB202190), RANKL + CK2.3, RANKL + CK2.3 + inhibitor (U0126, CAPE, or SB202190), RANKL + BMP2, RANKL + BMP2 + inhibitor (U0126, CAPE, or SB202190), untreated, and untreated + inhibitor (U0126, CAPE, or SB202190). The concentration of each reagent was as follows: RANKL, 10 ng/mL; CK2.3, 100 nM; BMP2, 40 nM; U0126, 1 µM; CAPE, 1 µM; and SB202190, 10 µM. All the inhibitors were purchased from Selleck Chemicals (Houston, TX, USA).

The medium was changed after 3 days, and fresh stimulation reagents were added again as indicated above. On Day 5, the cells were fixed with 4% paraformaldehyde for 20 min at room temperature and stained for tartrate resistant acid phosphatase (TRAP) using an Acid Phosphatase Leukocyte (TRAP) kit (Sigma-Aldrich, Cat# 387A-1KT). Osteoclasts were stained purple when observed under a brightfield microscope and counted in each treatment.

### 2.4. Immunofluorescence

Mouse femurs were fixed in 10% neutral buffered formalin for 48 h and decalcified in 14% ethylenediamine tetraacetic acid (EDTA) for 3–4 weeks. Chemical endpoints were tested with ammonium hydroxide/ammonium oxalate (1:1 *v*/*v*). Paraffin embedding and sectioning were performed by the Histochemistry and Tissue Processing Core at Nemours/Alfred I. DuPont Hospital for Children (Wilmington, DE, USA). Sectioned samples were deparaffinized in two changes of 100% xylene. Then, they were rehydrated in a series of 100% ethanol, 96% ethanol, and 70% ethanol for 5 min each, and deionized water for 30 s. Antigen retrieval was performed by incubation with testicular hyaluronidase (Sigma-Aldrich, Cat# H3884-100MG) at 37 °C for 30 min. Then, they were incubated in 3% BSA/0.1% Saponin for 1 h at room temperature to block nonspecific binding. p-Erk1/2 was fluorescently labeled with the primary p-Erk1/2 (mouse) antibody (Santa Cruz Biotechnology (Dallas, TX, USA), sc-7383) at a 1:100 dilution (in 1% BSA/0.1% saponin) for 1 h at room temperature. Additionally, a section sample designated as the “secondary control” was also incubated in 1% BSA/0.1% saponin. After 1 h of incubation in p-Erk1/2 antibody or 1% BSA/0.1% saponin, the section samples (including the secondary control sample) were incubated with the secondary antibody Alexa Fluor 488 donkey anti-mouse (Molecular Probes (Eugene, OR, USA), Cat# A21202) at a 1:1000 dilution (in 1% BSA/0.1% saponin) for 1 h at room temperature. Afterward, nuclei were stained with Hoechst 3342 (Fisher Scientific (Hampton, NH, USA), Cat# H1399) at a 1:5000 dilution (in deionized water) for 5 min. Another set of sectioned samples were labeled for Antennapedia Homeodomain (HD) using the same procedure as above. The primary antibody was HD antibody (mouse) (Developmental Studies Hybridoma Bank (Iowa City, IA, USA), Cat# 4C3) and used at a 1:100 dilution (in 1% BSA/0.1% Saponin), and the secondary antibody was Alexa Fluor 568 donkey anti-mouse (Invitrogen (Carlsbad, CA, USA), Cat#A10037) and used at a 1:1000 dilution (in 1% BSA/0.1% saponin). Finally, images were taken on a Zeiss Axiophot (Zeiss, Oberkochen, Germany) at 200× total magnification.

The following procedure was followed for the immunofluorescence of RAW264.7 cells. RAW264.7 cells were plated on coverslips in a 6-well plate at 1.8 × 10^4^ cell/cm^2^ in DMEM supplemented with 1.8 g/L of sodium bicarbonate, 2 mM l-Glutamine, 1 mM sodium pyruvate, and 100 U/L of penicillin and 0.1 mg/L of streptomycin. After plating, the cultured plate was kept in the incubator at 37 °C with 5% CO_2_ for 24 h. Then, the RAW264.7 cells were serum starved for 18–20 h before being treated with RANKL at 10 ng/mL, RANKL + CK2.3 at 100 nM, or RANKL + BMP2 at 40 nM. After 24 h of treatment, the coverslips were removed from the 6-well-plate and fixed/permeabilized in methanol/acetone for 10 min and 1 min, respectively, at −20 °C. Then, they were incubated in 3% BSA/0.1% saponin for 1 h at room temperature to block non-specific binding. p-Erk1/2 was fluorescently labeled with-p-Erk1/2 (rabbit) antibody (Cell Signaling (Danvers, MA, USA), Cat#9101) at a 1:100 dilution (in 1% BSA/0.1% saponin) for 1 h at room temperature. The secondary control coverslip was incubated in 1% BSA/0.1% saponin instead of the primary antibody. After 1 h of incubation in p-Erk1/2 antibody or 1% BSA/0.1% saponin, the coverslips were incubated with the secondary antibody Alexa Fluor 488 donkey anti-rabbit (Invitrogen, Cat# A21206) at a 1:1000 dilution (in 1% BSA/0.1% saponin) for 1 h at room temperature. Subsequentially, BMPRIa rabbit was fluorescently labeled with BMPRIa (rabbit) antibody (Santa Cruz Biotechnology, Cat# sc-20736) at a 1:100 dilution (in 1% BSA/0.1% Saponin) for 1 h at room temperature. The secondary control coverslip was incubated in 1% BSA/0.1% saponin. After 1 h of incubation in BMPRIa antibody or 1% BSA/0.1% saponin, the coverslips were incubated with the secondary antibody Alex Fluor 647 goat anti-rabbit (Life Technologies (Carlsbad, CA, USA), Cat#A21245) at a 1:1000 dilution (in 1% BSA/0.1% saponin) for 1 h at room temperature. Afterward, nuclei were stained with Hoechst 3342 at 1:5000 (in deionized water) for 5 min. Images were taken on an LSM710 confocal microscope (Zeiss, Oberkochen, Germany) at 400× total magnification.

The following procedure was followed for the immunofluorescence of primary osteoclasts. Femurs were isolated from 6-month-old B6 mice (donated by Dr. Lachke’s laboratory). Bone marrow was flushed out of the femurs and plated in alpha-Medium Essential Medium (α-MEM) (Calsson Labs (Smithfield, UT, USA), Cat# MEL08-500ML) (supplemented with 100 U/mL of penicillin, 100 µg/mL of streptomycin, 250 ng/mL of amphotericin B with 85 mg/L of NaCl, and 2 mM L-Gln) with 30 ng/mL of M-CSF (Novus Biologicals (Littleton, CO USA), Cat# NBP2-59586-50ug) overnight. The next day, non-adherent cells were collected and plated on coverslips in a 6-well-plate in the medium described above with the addition of 10 ng/mL of RANKL for 24 h. Afterward, p-ERK1/2 was fluorescently labeled with p-ERK1/2 (rabbit) antibody and Alexa Flour 488 donkey anti-rabbit using a protocol similar to that described above.

### 2.5. Image Analysis

Immunofluorescent images were analyzed by utilizing ImageJ (NIH, Bethesda). First, the images were converted into 8-bit image type, to allow the images’ thresholds to be adjusted. The threshold of the secondary control image was adjusted first. The upper threshold level was set at 255, while the lower threshold level was adjusted until the particle intensity was barely shown. Then, other images’ thresholds were adjusted using the same threshold levels used for the secondary control images. The original particle intensity of each image was measured with the “Histogram” function. Additionally, the particle intensity of a background area of each image was measured and subtracted from the original particle intensity of the same image to obtain the final particle intensity of that image.

### 2.6. Western Blot

RAW264.7 cells were plated on a 6-well plate at 1.8 × 10^4^ cell/cm^2^ in DMEM supplemented with 1.8 g/L of sodium bicarbonate, 2 mM l-Glutamine, 1 mM sodium pyruvate, and 100 U/L of penicillin and 0.1 mg/L of streptomycin. After plating, the cultured plate was kept in the incubator at 37 °C with 5% CO_2_ for 24 h. Then, RAW264.7 cells were serum starved for 18–20 h before being stimulated with RANKL at 10 ng/mL, RANKL + CK2.3 at 100 nM, or RANKL + BMP2 at 40 nM. After 24 h, 48 h, and 72 h after stimulation, the cells were lysed with RIPA lysis buffer supplemented with phenylmethylsulfonyl fluoride (PMSF), protease inhibitor, and phosphatase inhibitors, freshly added. The cells were lysed in lysis buffer for 1 h on ice while shaking. Cell debris was pelleted and discarded. The supernatant was recovered as the cell lysate.

The cell lysate was mixed with Laemmli sample buffer to a 1× final concentration and heated up to 95 °C for 5 min. Then, protein separation was performed on a 10% polyacrylamide gel at 90 V for 1.5 h. After the separation, the proteins were transferred from the gel to a PDVF membrane by blotting using the Trans-Blot SD Semi-Dry Transfer Cell (Bio-Rad, Hercules, CA, USA) at 15 V for 50 min. Then, the blot was incubated in 3% BSA for 1 h at room temperature on a shaker to block non-specific binding. Phospho Erk1/2, BMPRIa, and GAPDH were detected in the cell lysates by incubation with anti-phospho Erk1/2 (rabbit), anti-BMPRIa (rabbit), and anti-GAPDH (mouse) antibodies (Novus Biologicals, Cat#NB300) at 1:1000 dilutions (in 1% BSA) overnight at 4 °C on a shaker. This was followed by incubation with the secondary antibody HRP anti-rabbit at a 1:5000 dilution (in 1% BSA). Lastly, the blot was incubated in Chemiluminescent FemtoMax Super Sensitive HRP Substrate (Rockland (Pottstown, PA, USA), Cat# Femtomax-110) for 2 min. Protein bands were detected with a Chemidoc Imaging System (Bio-Rad, Hercules, CA, USA).

### 2.7. Primary Preosteoclast Isolation and Differentiation

Spleens were isolated from mice 2 weeks after the first injection. Primary cells were flushed out of the spleens with primary culture medium (Alpha Minimum Essential Medium (a-MEM) supplemented with 10% FBS, 100 U/L of penicillin, and 0.1 mg/L of streptomycin). Then, he cells were centrifuged and pelleted. The cell pellet was resuspended in primary culture medium with 50 ng/mL of RANKL and 30 ng/mL of M-CSF for 7 days. RNA extraction and real-time polymerase chain reaction (RT-PCR) then followed.

### 2.8. RNA Extraction and Real-Time Polymerase Chain Reaction (RT-PCR)

Cells were lysed with TRIzol Reagent (Life Technologies, Cat# 15596018) at 1 mL per 1 × 10^6^ cells for 5 min at room temperature. Then, chloroform was added at 0.2 mL per 1 mL of TRIzol used, and the mixture was shaken for 15 s, followed by incubation for 3 min at room temperature, then centrifugation at 12,000× *g* for 15 min at 4 °C. The clear top aqueous layer that contained the RNA was collected. Isopropanol was added to the collected aqueous layer at 0.5 mL per 1 mL of TRIzol used then vortexed, followed by incubation for 10 min at room temperature, then centrifugation at 12,000× *g* for 10 min. The pellet was collected and suspended in 75% ethanol at 1 mL per 1 mL of TRIzol used then vortexed. Centrifugation was performed at 7500× *g* for 5 min at 4 °C to collect the pellet. After air drying, the RNA pellet was resuspended in nuclease-free water.

Two-step RT-PCR was performed to examine the expression of osteoclast marker genes. cDNA synthesis was performed using the ImProm-II Reverse Transcription System (Promega (Madison, WI, USA), Cat#A3800) according to manufacturer’s protocol. RT-PCR was performed using the Fast SYBR Green Master Mix (Applied Biosystems (Foster City, CA, USA), Cat# 4385612) according to the manufacturer’s protocol. The primers for osteoclast marker genes and GAPDH are listed in Table 1.

### 2.9. Statistical Analysis

The results are shown as the mean ± standard error (STE). Statistical analysis was performed using One-Way ANOVA followed by Fisher’s Least Significant Difference post-hoc test. From left to right, each bar on the graph is ordered alphabetically (a, b, c, …), and (a) denotes a significant difference from the first bar, while (b) denotes a significant difference from the second bar, and so on.

## 3. Results

### 3.1. Effect of CK2.3 Concentration on Viability of RAW264.7 Cells

A concentration curve of CK2.3 was generated to determine the concentration at which CK2.3 would not affect the viability of RAW264.7 cells during RANKL-induced osteoclast development (Figure 2). We observed no changes in cell population between control and RANKL stimulation. At a 50 nM concentration of CK2.3, a decrease in the cell population was observed. However, at a 500 nM concentration of CK2.3, we observed an increase in the cell population. Additionally, there were no changes in the cell population at 100 nM and 1000 nM concentrations of CK2.3.

### 3.2. MEK Inhibitor U0126 Abolished the Effect of CK2.3 as an Antagonist of RANKL-Induced Osteoclastogenesis of RAW264.7 Cells

We wanted to identify the potential signaling pathway(s) that are involved in CK2.3’s suppression of the RANKL-induced osteoclast development of RAW264.7 cells. We previously showed that CK2.3 activated the ERK signaling pathway in C2C12 cells during osteoblast development [53]. Additionally, the p38 and NF-κB signaling pathways have been shown to be important for osteoclastogenesis for the development of the skeletal system [44,50]. Thus, we investigated the effect of the MEK inhibitor U0126, p38 inhibitor SB202190, and NF-κB inhibitor CAPE on the antagonistic effect of CK2.3 on RANKL-induced osteoclastogenesis (Figure 3, Figure 4 and Figure 5, respectively). First, we generated a concentration curve for each inhibitor vs. cell viability to determine the working concentration of each inhibitor (Figure 3A, Figure 4A, and Figure 5A). The highest concentration at which the cell viability was not affected was chosen. We observed that the MEK inhibitor U0126 abolished the effect of CK2.3 as an antagonist of the RANKL-induced osteoclastogenesis of RAW264.7 cells (Figure 3B). U0126 restored the RANKL-induced osteoclastogenesis of RAW264.7 cells that were cocultured with CK2.3 (Figure 3B). The p38 inhibitor SB202190 and NF-κB inhibitor CAPE, on the other hand, showed no effect on CK2.3’s ability as an antagonist of the RANKL-induced osteoclastogenesis of RAW264.7 cells (Figure 4B and Figure 5B, respectively).

### 3.3. CK2.3 Induced ERK Activation in RAW264.7 Cells and Primary Osteoclasts 24 h after CK2.3 Stimulation

Since the MEK inhibitor U0126 nullified the effect of CK2.3 of suppressing the RANKL-induced osteoclast development of RAW264.7 cells, we wanted to confirm that CK2.3 regulated ERK activation, a downstream target of MEK, in RAW264.7 cells. We observed that CK2.3 increased the phosphorylation of Erk1/2 (p-Erk1/2) after 24 h of stimulation in RAW264.7 cells, as determined by Western blotting (Figure 6A) and immunofluorescence (Figure 6B). Furthermore, a CK2.3-induced increase in p-Erk1/2 was verified in primary osteoclasts isolated from 6-month-old mice, as determined by immunofluorescence (Figure 6C).

### 3.4. CK2.3 Upregulated BMPRIa Expression in RAW264.7 Cells after 24 h of CK2.3 Stimulation

We designed CK2.3 as an activator of the BMP2 signaling pathway. Since it was observed that CK2.3 increased p-Erk1/2 after 24 h of stimulation with CK2.3 (Figure 7), we investigated if CK2.3 increased p-Erk1/2 through the BMP2 signaling pathway. BMP2 signals through BMPRIa to mediate downstream signaling pathways such as ERK [54]. It was shown that BMPRIa was also upregulated after 24 h of stimulation with CK2.3 in RAW264.7 cells, as determined by Western blotting (Figure 7A) and immunofluorescence (Figure 7B).

### 3.5. CK2.3 Did Not Mediate the Canonical SMAD Signaling Pathway of BMP2 Signaling Transduction in RAW264.7 Cells

The observation of the upregulation of BMPRIa by CK2.3 led us to investigate if the canonical SMAD signaling pathway was also upregulated in CK2.3-mediated BMPRIa expression in RAW264.7 cells. SMAD signaling has been reported to be involved in osteoclast development [45]. However, no changes were observed in the phosphorylation of Smad1/5 (p-Smad1/5) at either 24 h or 48 h after stimulation with CK2.3, as determined by Western blotting (Figure 8).

### 3.6. CK2.3 Downregulated Osteoclast Marker Genes in Primary Preosteoclasts Isolated from the Spleens of 6-Month-Old Retired Breeder Mice

To demonstrate the effect of CK2.3 in suppressing osteoclast development ex vivo, we isolated primary preosteoclasts from the spleens of CK2.3-injected 6-month-old retired breeder mice and cultured them in RANKL for 7 days. We previously reported that the systemic injection of CK2.3 into these mice improved their bone formation [41]. Here, we observed that osteoclast markers such as ATP6v6d02, TRAP, Cstk, DC-STAMP, and NFATc1 decreased at Week 2 at all three different concentrations (Figure 9).

### 3.7. CK2.3 Uptake into the Femurs of 6-Month-Old Retired Breeder Mice Was Detected at Week 2 after Injection Followed by Upregulation of p-Erk1/2 in the Femur

Because it was observed that osteoclast marker genes were decreased in RANKL-induced primary osteoclasts isolated from the spleen ex vivo at Week 2 after CK2.3 injection, we sought to investigate whether the presence of CK2.3 was still detectable at Week 2 in the bone. CK2.3 peptide was designed to contain an antennapedia homeodomain to facilitate the cellular uptake of CK2.3 [55]. At the same time, we could also trace CK2.3 in the cell with an antibody against the antennapedia homeodomain. It was shown that CK2.3 uptake was detected in the femur at Week 2 after the injection (Figure 10A). Additionally, it was observed that p-Erk1/2 was increased in the femur at Week 2, as determined by immunofluorescence (Figure 10B).

## 4. Discussion

Bone remodeling is a dynamic process and important for the development and maintenance of the skeletal system. However, aging tends to decrease the efficiency of the bone remodeling process, which results in the decline of bone mass [24]. Osteoporosis is a disease characterized by significant bone loss. Osteoporosis typically occurs in the population aged 50 or older. On the other hand, idiopathic juvenile osteoporosis has an earlier onset at childhood and interferes with the development of the skeletal system. Regardless of the age of onset, osteoporotic patients have low bone mineral density and are at high risk of experiencing bone fractures. Our laboratory designed a novel peptide, CK2.3, as a novel treatment for osteoporosis [39,40,41,42]. CK2.3 induced bone formation through the BMP2 signaling pathway [39,40]. CK2.3 induced bone formation by enhancing osteoblast development while suppressing osteoclast development [41,42]. It was shown that CK2.3 enhanced osteoblastogenesis through activating the ERK signaling pathway [41,53]. In this study, we demonstrated that CK2.3 also acted through the ERK signaling pathway to mediate the suppression of osteoclast development. CK2.3 activated the ERK signaling pathway both in vitro (Figure 6) and in vivo (Figure 10) and downregulated osteoclast marker genes such as ATP6v6d02, TRAP, Cstk, DC-STAMP, and NFATc1 ex vivo (Figure 9).

It has been shown that p38 MAPK, ERK MAPK, and NF-κB play an important role in RANKL-induced osteoclast development [44,45,46,51]. RANK is a member of the TNF family. The signal transduction of RANK begins with the recruitment of TRAF adaptor proteins to the cytoplasmic tail of RANK. TRAF1, TRAF2, TRAF3, TRAF5, and TRAF6 were identified as associating with RANK [11]. TRAF6 was shown to be important in osteoclastogenesis during skeletal system development. TRAF6^−/−^ mice developed osteopetrosis [56,57]. TAK1 is an important inducer of MAPK and NF-κB [58,59,60,61,62]. Two important TAK1 adaptor proteins are TAB1 and TAB2. TAB1 associates with TAK1 within 68 C-terminal amino acids and is an activator of TAK1 via the autophosphorylation of threonine/serine residues in the activation loop [63,64]. Finally, TAB2 links TRAF6 and the TAK1/TAB1 complex to induce a signal transduction cascade [20,65]. Additionally, osteoclastogenesis is also regulated by BMP2 [45,66,67,68,69,70]. BMP2 is reported to upregulate p-p38 during Days 1–2 of RANKL-induced osteoclast development but not p-ERK [45]. BMP2 binds BMPRIa to recruit BMPRII to form a BMPRIa–BMPRII complex [71]. BMPRII phosphorylates BMPRIa to activate the SMAD canonical and MAPK non-canonical pathways, which have been suggested to be expressed at distinct points in time during osteoclast development [45]. In the BMP signaling pathway, X-linked inhibitor of apoptosis (XIAP) is identified as an adaptor that links TAB1/TAK1 to BMPRIa [72]. XIAP mediates BMPRIa and TAB1/TAK1 coupling through a ring zinc finger domain and BIR1 motif that associate with BMPRIa and TAB1/TAK1, respectively [72,73]. The association of the TAB1/TAK1/XIAP complex with neurotrophin receptor interacting MAGE (NRAGE) is shown to be important for the activation of MAPK [54]. Downstream of TAK1 are MAPKK family members such as MEK1/2 and MKK3/6 [63].

In this study, it was shown that the MEK inhibitor U0126 nullified the effect of CK2.3 as an antagonist of osteoclast development and restored the RANKL-induced osteoclast development of RAW264.7 cells, while the p38 inhibitor SB202190 and NF-κB inhibitor CAPE did not (Figure 4 and Figure 5, respectively). In the presence of RANKL alone, however, U0126 inhibited the osteoclast development (Figure 3). Thus, it was shown that the activation of ERK was still important for the osteoclast development of RAW264.7 cells [45,46,47,48,49]. It was speculated that ERK played a dual role in the regulation of osteoclast development. Looking at the time-point activation of ERK, it was observed that the activation of ERK MAPK by CK2.3 increased 24 h after stimulation with CK2.3 in RAW264.7 cells and primary osteoclasts (Figure 6). It has been shown that the opposite effect of estrogens on the anti-apoptotic and pro-apoptotic pathways of osteoclasts is due to the duration of ERK MAPK activation by estrogens [74]. Estrogens have been used in hormone replacement therapy to increase bone mass to treat osteoporosis [75]. It has been shown that transient ERK MAPK activation (5–30 min) abolishes the anti-apoptotic effect of estrogens on osteoclasts. However, sustained ERK activation for at least 24 h results in a pro-apoptotic effect of estrogens on osteoclasts. These results demonstrate a dual role for ERK MAPK in the regulation of osteoclast development depending on the transient or sustained activation of ERK MAPK. It is shown that during RANKL-induced osteoclast development, ERK is transiently (10 min) activated [49]. Together, a hypothetical role of CK2.3 in which the suppression of osteoclastogenesis by CK2.3 is driven by the activation of ERK MAPK at 24 h after stimulation with CK2.3 is suggested.

Targeting the ERK MAPK signaling pathway has been employed as a therapeutic strategy to treat cancer and neurological diseases, highlighting its importance in the regulation of these diseases [76,77]. Similarly, although there has yet to be an available therapeutic strategy that is designed to specifically target the ERK MAPK signaling pathway to treat osteoporosis, various other strategies such as estrogen, rhein-derived thioamide (RT), prostaglandin E2 (PGE2), and Src inhibitors, to name a few, have mechanisms involved in the regulation of the ERK MAPK signaling pathway [74,78,79,80,81,82]. Estrogen and PGE2 activate ERK to promote osteoblast differentiation, but estrogen also activates ERK to inhibit osteoclast differentiation, as mentioned above [74,81]. RT, on the other hand, inhibits the activation of ERK, which inhibits osteoclast differentiation. Src is an important component of the complex that bridges RANK and TAK1, resulting in the activation of downstream signaling including that of ERK MAPK. Thus, Src inhibitors can potentially act as an antagonist of ERK MAPK activation in osteoclast differentiation. While RT and PGE2 are still at an investigative stage, estrogen and the Src inhibitor Saracatinib (AZD0530) have been approved by the FDA and reached the clinical trial stage, respectively [83]. Thus, together with our findings of the involvement of CK2.3 in the activation of ERK MAPK to promote osteoblastogenesis and to inhibit osteoclastogenesis, it is shown that targeting the ERK MAPK signaling pathway could potentially lead to a novel therapeutic strategy to treat osteoporosis

CK2.3 was designed as a CK2 blocking peptide that mediated BMP2 signaling transduction [39,40,41,43]. Since it was shown that ERK MAPK was activated at 24 h after stimulation with CK2.3 in RANKL-induced osteoclast development (Figure 6), it was not a surprise that an increase in BMPRIa expression was observed 24 h after CK2.3 stimulation (Figure 7). BMP2 has been reported to play a positive role in osteoclast differentiation [45,67,69,70,84]. However, the expression BMPRIa might play a different role rather than enhancing osteoclast differentiation. It is shown that the disruption of BMPRIa in osteoclasts increased osteoclast formation but resulted in smaller and fewer nuclei, suggesting that BMPRIa played a role in osteoclast fusion [85]. In fact, it has been shown that the knockout of BMPRII decreases osteoclast differentiation [45]. These findings suggest that BMPRII, rather than BMPRIa, is required for osteoclast differentiation. It has been reported that BMPRII expression is gradually increased from Day 1–5 in RANKL-induced osteoclast differentiation, while BMPRIa expression only begins to be detected from Day 2–5 [69]. Thus, we speculated that BMPRII bound to a partner other than BMPRIa to induce the commitment of osteoclast precursors to osteoclasts on Day 1. BMP receptor type Ib (BMPRIb) was shown to be constantly expressed from Day 0–5 in RANKL-induced osteoclast differentiation and had a weaker affinity (3-fold weaker) for BMP2 than BMPRIa [69,86]. Additionally, soluble forms of BMPRIb show better inhibitory effects on osteoclast formation in vivo than soluble forms of BMPRIa [87]. Thus, it was speculated that BMPRII bound to BMPRIb on Day 1 to induce the commitment of osteoclast precursors into osteoclasts. With the expression of BMPRIa on Day 2, BMPRII switched partners to induce the fusion of osteoclasts. Since CK2.3 induced the expression of BMPRIa on Day 1 to make it a preferred partner of BMPRII, the osteoclast-commitment signal transduction was, therefore, inhibited.

With an increased expression of BMPRIa 24 h after CK2.3 expression (Figure 7), it was speculated that the canonical SMAD signaling pathway was also activated. However, no changes in p-Smad1/5 expression were detected (Figure 8). Thus, it was suggested that CK2.3’s mechanism of action in inhibiting osteoclast differentiation did not involve the SMAD canonical signaling pathway but the non-canonical ERK MAPK signaling pathway. This showed that CK2.3 and BMP2 affected the BMP2 signaling transduction differently. It has been reported that in BMP signaling transduction, the choice of whether the SMAD or non-SMAD signaling pathway should be activated depends on the way in which BMP binds to the receptor complex. If BMP binds to a preformed BMP receptor type I/BMP receptor type II complex, the SMAD signaling pathway is expressed [88]. On the other hand, if BMP binds to BMP receptor type 1 or BMP receptor type II to recruit the other, the non-SMAD signaling pathway is expressed [71,89]. It is reported that BMP2 binds to BMPRIa to recruit BMPRII, which leads to caveola-mediated internalization and the activation of the non-SMAD signaling pathway [89]. We previously reported that CK2.3 internalization occurred through caveola-mediated endocytosis [90]. Thus, we speculated that CK2.3’s selective activation of the ERK MAPK rather than the SMAD signaling pathway was also due to CK2.3’s ability to mediate caveolar endocytosis.

CK2.3 was designed to contain an Antennapedia homeodomain to help with cellular uptake [55]. Additionally, it could also be used to detect the presence of CK2.3 in the cells. The ex vivo effect of CK2.3 was shown by a decreased expression of osteoclast marker genes such as ATP6v6d02, TRAP, Cstk, DC-STAMP, and NFATc1 at Week 2 after the first injection of CK2.3 in 6-month-old retired breeder mice (Figure 9). Furthermore, the presence of CK2.3 was also detected in the femurs of these mice, followed by an increase in p-Erk1/2 (Figure 10). It should be noted that in this study, each injection of CK2.3 was administered at a 2.3 µg/kg concentration per mouse, compared to commonly used anti-resorptive bisphosphonates, which are often used at higher concentrations (60–500 µg/kg) to show a positive effect in mice [91].

Recent studies demonstrate that CK2 interacting protein-1 (CKIP-1) plays an important role in the progression of osteoporosis [92]. Additionally, the CK2 inhibitor CX-4945 has been shown to promote osteoblastogenesis and inhibit osteoclastogenesis [93]. We designed our peptide CK2.3 as a CK2 binding peptide, and its function in the regulation of osteoblast and osteoclast differentiation was similar to that of CX-4945 [40,41,42]. Thus, another hypothetical role of CK2.3 could be as an inhibitor of CK2. This, in turn, could affect the activity of CKIP-1 and slow down the progression of osteoporosis. In this study, we also demonstrated that CK2.3 and BMP2 affected the BMP2 signaling pathway differently. Clinical evidence shows that rhBMP2 is less successful in the clinic due to a variety of side effects and complications [94,95]. Furthermore, several researchers have reported unresponsiveness to the BMP2 stimulation of bone marrow stromal cells (BMSCs) isolated from osteoporotic patients [96,97,98]. These reports may be related to reports on the aberrance of BMP2 signaling in osteoporotic patients [96,97,98,99,100,101,102,103]. We have also reported the unresponsiveness of BMSCs isolated from osteoporotic patients to BMP2, but they respond to CK2.3 [104]. Thus, the different effect of CK2.3 on the BMP2 signaling pathway compared to that of BMP2 might also contribute to the successful response of osteoporotic BMSCs to CK2.3 [104].

## 5. Conclusions

This study demonstrated the mechanism by which CK2.3 regulated osteoclastogenesis by multiple mechanisms. CK2.3 inhibited the commitment of osteoclast precursors to osteoclasts, with the early expression of BMPRIa on Day 1. Additionally, the upregulation of the ERK MAPK signaling pathway resulted from the expression of BMPRIa at Day 1, the timepoint at which it has a negative impact on osteoclast differentiation. Another important discovery of this study was that it showed that CK2.3 activated different signaling pathways downstream of BMP2 signaling transduction in osteoblasts and osteoclasts (Figure 11). Understanding more about the mechanism of CK2.3 as an antiresorptive agent will allow CK2.3 to be developed as a better and safer treatment to improve bone formation to combat diseases such as idiopathic juvenile osteoporosis, which hinders bone development in childhood.

## Figures and Tables

**Figure 1 jdb-08-00012-f001:**
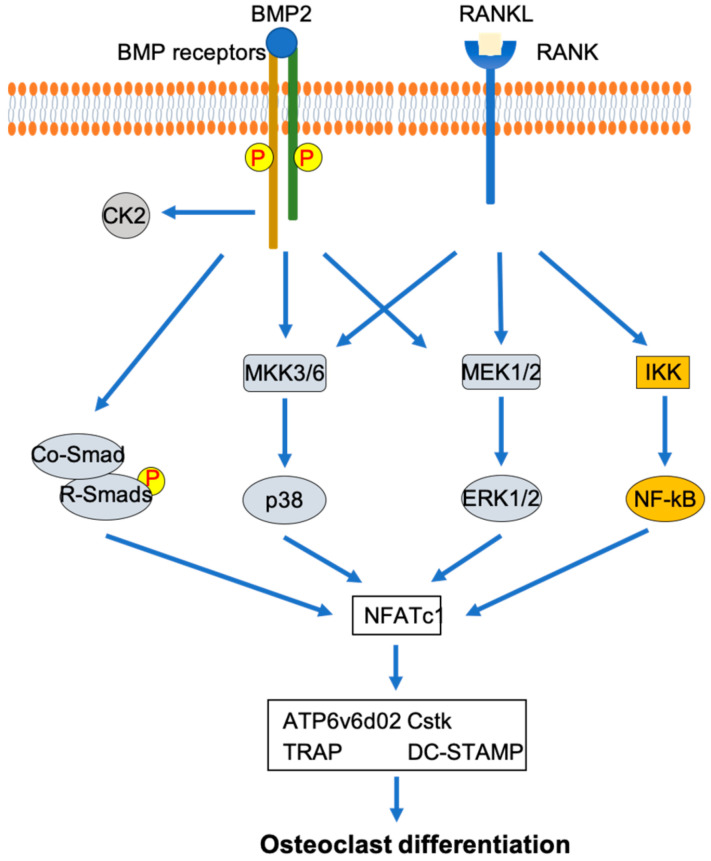
Involvement of p38 MAPK, ERK MAPK, and NF-κB signaling pathways in osteoclast differentiation mediated by RANKL and BMP2.

**Figure 2 jdb-08-00012-f002:**
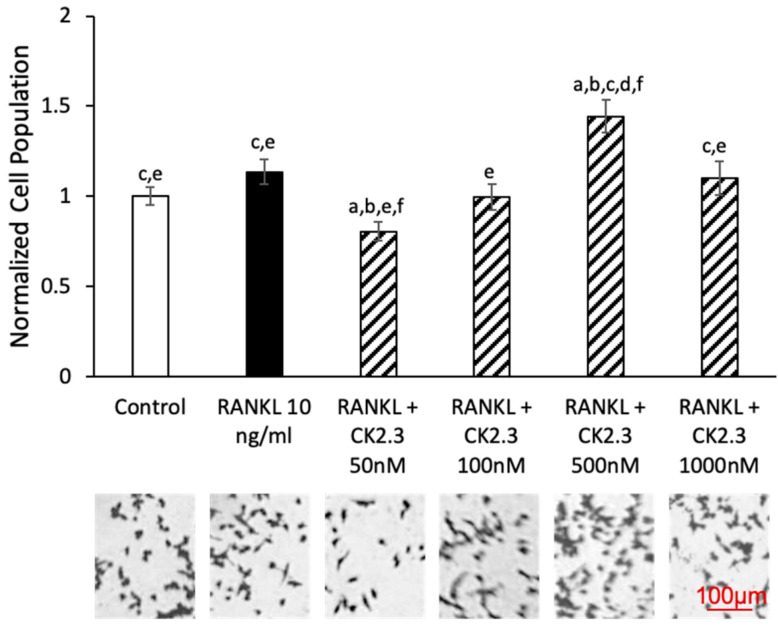
Effect of CK2.3 on RAW264.7 cell growth. RAW264.7 cells were stimulated with different concentrations of CK2.3 for 5 days. Cell growth was affected by CK2.3 at 50 nM and 500 nM. However, it was not affected at 100 nM and 1000 nM. a: denotes significant differences from Control, b: denotes significant difference from RANKL at 10 ng/mL, c: denotes significant difference from RANKL + CK2.3 at 50 nM, d: denotes significant difference from RANKL + CK2.3 at 100 nM, e: denotes significant difference from RANKL + CK2.3 at 500 nM, and f: denotes significant difference from RANKL + CK2.3 at 1000 nM.

**Figure 3 jdb-08-00012-f003:**
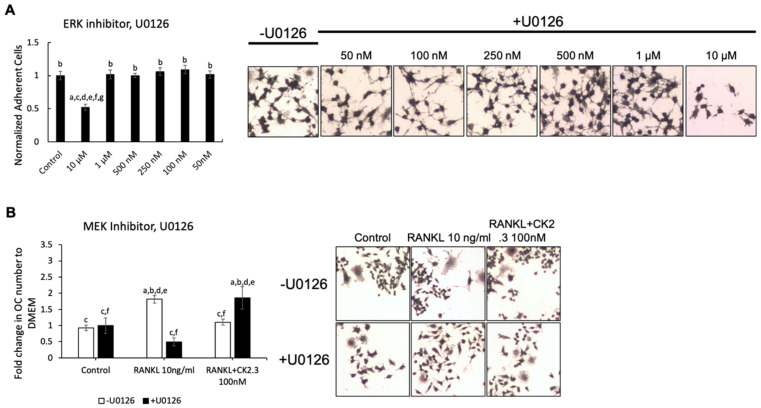
MEK inhibitor, U0126, abolished the antagonistic effect of CK2.3 on RANKL-induced osteoclastogenesis. (**A**) Concentration curve for U0126 to determine the working concentration. a: denotes significant difference from Control, b: denotes significant difference from 10 µM, c: denotes significant difference from 1 µM, d: denotes significant difference from 500 nM, e: denotes significant difference from 250 nM, f: denotes significant difference from 100 nM, and g: denotes significant difference from 50 nM. (**B**) The effect of U0126 on the antagonistic effect of CK2.3 on RANKL-induced osteoclastogenesis. a: denotes significant difference from Control without inhibitor, b: denotes significant difference from Control with inhibitor, c: denotes significant difference from RANKL at 10 ng/mL without inhibitor, d: denotes significant difference from RANKL at 10 ng/mL with inhibitor, e: denotes significant difference from RANKL + CK2.3 at 100 nM without inhibitor, and f: denotes significant difference from RANKL + CK2.3 at 100 nM with inhibitor.

**Figure 4 jdb-08-00012-f004:**
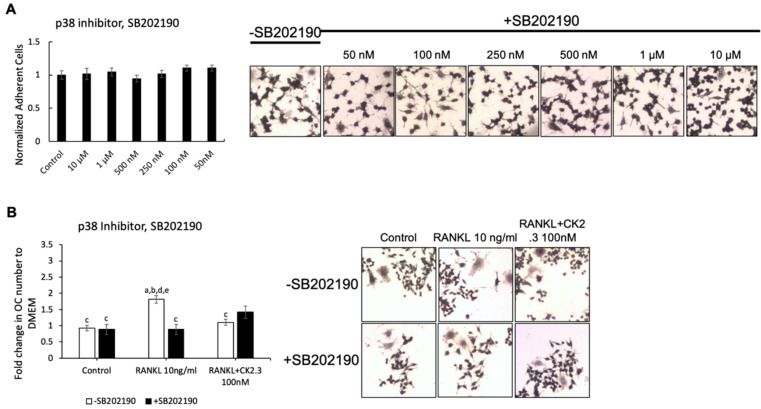
p38 inhibitor, SB202190, did not affect the antagonistic effect of CK2.3 on RANKL-induced osteoclastogenesis. (**A**) Concentration curve for SB202190 to determine the working concentration. (**B**) The effect of SB202190 on the antagonistic effect of CK2.3 on RANKL-induced osteoclastogenesis. a: denotes significant difference from Control without inhibitor, b: denotes significant difference from Control with inhibitor, c: denotes significant difference from RANKL at 10 ng/mL without inhibitor, d: denotes significant difference from RANKL at 10 ng/mL with inhibitor, and e: denotes significant difference from RANKL + CK2.3 at 100 nM without inhibitor.

**Figure 5 jdb-08-00012-f005:**
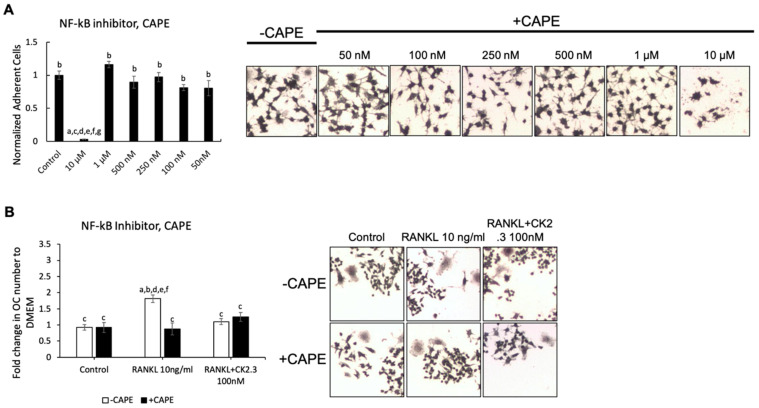
NF-κB inhibitor, caffeic acid phenethyl ester (CAPE), did not affect the antagonistic effect of CK2.3 on RANKL-induced osteoclastogenesis. (**A**) Concentration curve for CAPE to determine the working concentration. a: denotes significant difference from Control, b: denotes significant difference from 10 µM, c: denotes significant difference from 1 µM, d: denotes significant difference from 500 nM, e: denotes significant difference from 250 nM, f: denotes significant difference from 100 nM, and g: denotes significant difference from 50 nM. (**B**) The effect of CAPE on the antagonistic effect of CK2.3 on RANKL-induced osteoclastogenesis. a: denotes significant difference from Control without inhibitor, b: denotes significant difference from Control with inhibitor, c: denotes significant difference from RANKL at 10 ng/mL without inhibitor, d: denotes significant difference from RANKL at 10 ng/mL with inhibitor, and e: denotes significant difference from RANKL + CK2.3 at 100 nM without inhibitor.

**Figure 6 jdb-08-00012-f006:**
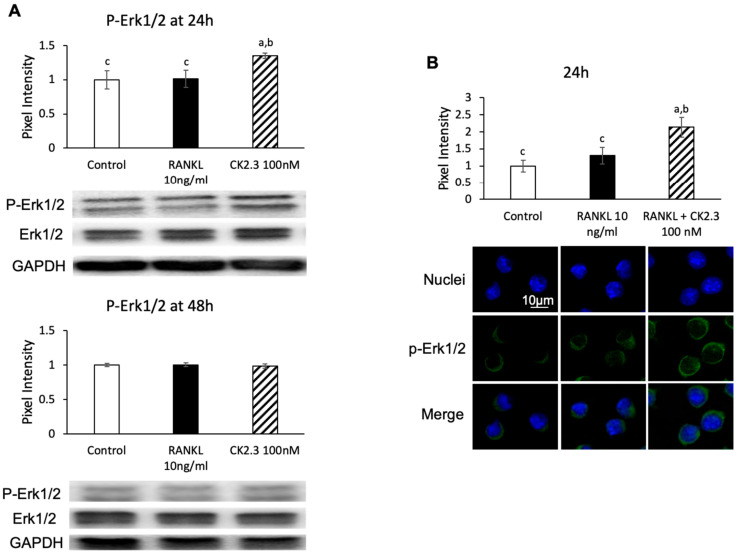
Effect of CK2.3 on p-Erk 1/2 in RANKL-induced osteoclastogenesis. CK2.3 increased p-Erk 1/2 after 24 h of stimulation in RAW264.7 cells, as determined by (**A**) Western blotting and (**B**) immunofluorescence. p-Erk 1/2 was increased after 24 h of stimulation, as determined by immunofluorescence in (**C**) primary osteoclast precursors of 6-month-old B6 mice. a: denotes significant difference from Control, b: denotes significant difference from RANKL at 10 ng/mL, and c: denotes significant difference from RANKL + CK2.3 at 100 nM.

**Figure 7 jdb-08-00012-f007:**
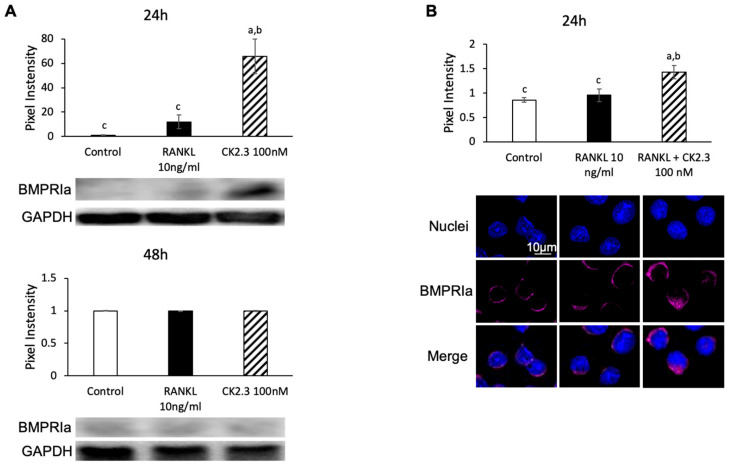
Regulation of BMPRIa expression by CK2.3. Expression of BMPRIa was upregulated in RAW264.7 cells after 24 h of CK2.3 stimulation, as determined by (**A**) Western blotting and (**B**) immunofluorescence. a: denotes significant difference from Control, b: denotes significant difference from RANKL at 10 ng/mL, and c: denotes significant difference from RANKL + CK2.3 at 100 nM.

**Figure 8 jdb-08-00012-f008:**
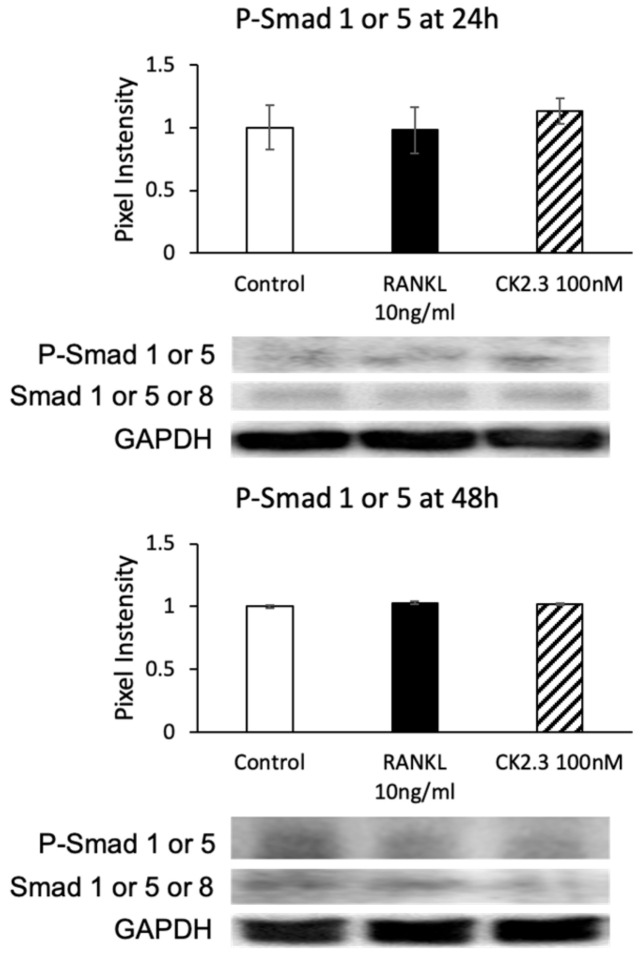
CK2.3 did not mediate the SMAD canonical pathway downstream of the BMP signaling pathway. The phosphorylation of Smad1 or 5 was not affected by CK2.3 after either 24 h or 48 h of stimulation in RAW264.7 cells.

**Figure 9 jdb-08-00012-f009:**
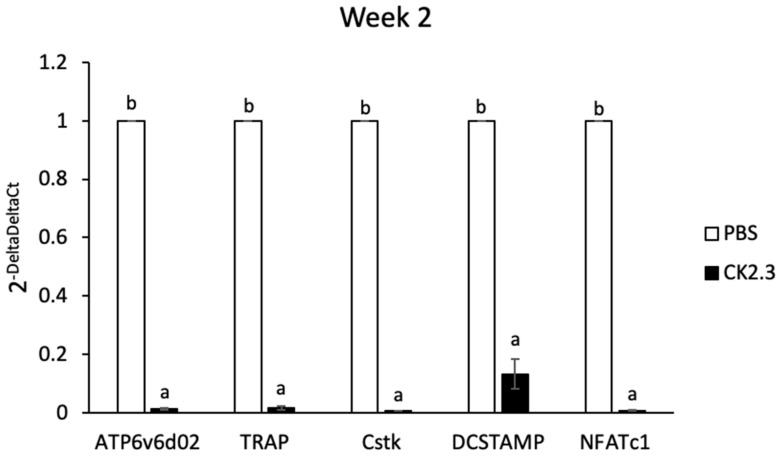
Analysis of osteoclast markers in primary preosteoclasts isolated from the spleens of 6-month-old retired breeder mice by RT-PCR. Primary preosteoclasts were differentiated to osteoclasts with RANKL for 7 days. Osteoclast markers were downregulated at Week 2 after the first injection of CK2.3. a: denotes significant difference from PBS, and b: denotes significant difference from CK2.3.

**Figure 10 jdb-08-00012-f010:**
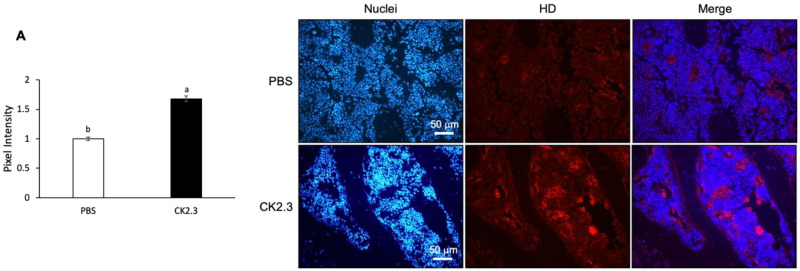
Uptake period and regulation of p-Erk1/2 by CK2.3 in the femurs of 6-month-old retired breeder mice. (**A**) 2 weeks after the injection, CK2.3 uptake was detected in the femurs of 6-month-old retired mice by immunofluorescence. (**B**) Increased p-Erk1/2 was detected by immunofluorescence in the femurs of 6-month-old retired breeder mice at Week 2 after the injection. a: denotes significant difference from PBS, and b: denotes significant difference from CK2.3 100.

**Figure 11 jdb-08-00012-f011:**
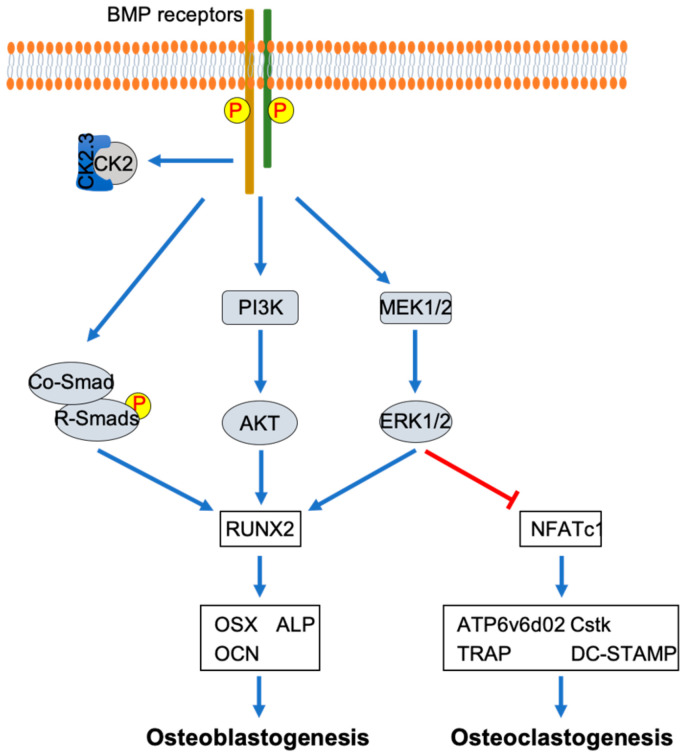
Differential mechanisms of CK2.3 in osteoblastogenesis and osteoclastogenesis. CK2.3 activates SMAD, AKT, and ERK, resulting in the upregulation of the osteoblastic master transcription factor RUNX2, which promotes osteoblastogenesis. On the other hand, CK2.3 activates ERK, resulting in the downregulation of the master transcription factor NFATc1, which inhibits osteoclastogenesis.

**Table 1 jdb-08-00012-t001:** Primers for osteoclast marker genes and GAPDH.

Gene	Reverse/Forward	Sequence
*ATP6v6d02*	Reverse	GTG CCA AAT GAG TTC AGA GTG ATG
	Forward	TCA GAT CTC TTC AAG GCT GTG CTG
*NFATc1*	Reverse	CGT ATG GAC CAG AAT GTG ACG G
	Forward	GGT GCC TTT TGC GAGCAG TAT C
*Cstk*	Reverse	GCT GGC TGG AAT CAC ATC TT
	Forward	AGG GAA GCA AGC ACT GGA TA
*TRAP*	Reverse	GAG TTG CCA CAC AGC ATC AC
	Forward	CGT CTC TGC ACA GAT TGC A
*BMPRIa*	Reverse	TGA GTC CAG GAA CCA GTG CCT TT
	Forward	CAG AAT CTA GAT AGT ATG C
*DC-STAMP*	Reverse	TGG CAG GAT CCA GTA AAA GG
	Forward	GGG CAC CAG TAT TTT CCT GA
*GAPDH*	Reverse	TAC TCA GCA CCA GCA TCA CC
	Forward	TGA CCC CTT CAT TGA CCT TC

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
