# Peer review of "A Synthetic Peptide, CK2.3, Inhibits RANKL-Induced Osteoclastogenesis through BMPRIa and ERK Signaling Pathway"

_jdb, 2020, doi:10.3390/jdb8030012_

Round 1
Reviewer 1 Report
The manuscript by Anja Nohe and colleagues previously reported casein kinase 2 (CK2) as an activator of BMP2 signal transduction. This Delaware study further demonstrates CK2.3 synthetic peptide inhibition of RANKL induced osteoclast differentiation through BMP receptor type Ia (BMPRIa) and ERK signaling pathway, which can be developed as a new therapy against osteoporosis. The paper is well written with a thorough background literature and conclusive of the results. I suggest the following minor comments and clarifications.
“Title- The authors may clarify the title for the words….RANKL induced osteoclastogenesis…” and in the text where appropriately used.
Fig.1 schema-I suggest showing RNKL binding to RANK receptors like BMP2 they have shown and clarify the word in the box shown as “DC-STAMP” as noted in the text. NATc1 is shown as the master regulator of genes underneath shown. Otherwise, they may show Fig.1 illustration in discussion section summarizing the results with signal molecules/gene transcription that CK2.3 upregulate or down-regulate with markings [upright or down arrows/inhibition (X)].
MEK inhibitor nullified the effect of CK2.3 on suppressing RANKL induced osteoclastogenesis and CK2.3 stimulate ERK phosphorylation (Fig.5A). The authors may specify CK2.3 is agonist or antagonist of osteoclast formation in the discussion or illustration figure reflecting the Title.
Fig.5C-Osteoclast is multinucleated. Please clarify the legend for osteoclast precursor or monocyte/macrophage cell types used to reflect the confocal microscopy shown.
Fig.7-clarify the figure labeling and legend for P-Smad 1 or 5 shown in upper and lower panels. Improve the contrast/quality of P-Smad shown in the upper panel.
Reviewer 2 Report
Dear Authors,
the manuscript "A Synthetic Peptide, CK2.3, Inhibits RANKL-Mediated Osteoclastogenesis through BMPRIa and ERK Signaling Pathway" started from the evidences obtained from previous study. The authors observed the ability to CK2.3 to mediated the mineralization and osteoblast development through the SMAD, ERK, and AKT signaling modulation, while in this study, understanding the role of CK2.3 in the regulation or modulation of osteblasts-osteoclasts balance.
The scientific interest of their investigation is significant; but according to me, the manuscript can be considered acceptable for the publications after a major revision.
In particular, to further improve the quality of the manuscript, the authors should discuss better the following points:
The role of ERK as inductor/modulator/regulator of bone cells activation and phenotypes is investigated in a different cell models, for this reason I suggest improving the references in introduction and discussion sections;
In the introduction section, I suggest to add clinical evidences about a therapeutic approaches that targetting ERK MAPK signaling, in order to underline the importance of their study.
In material and methods section, I suggest to add the better description of materials, in terms of: colture conditions, antibodies used, statistical analysis and details of the hystograms of the figures and figures legends, ec;
To revise the point "3.2. MEK inhibitor U0126 nullified the effect of CK2.3 on suppressing RANKL-mediated osteoclastogenesis of RAW264.7 cells", because there are mistakes in the description of the figures 3 and 4 results. In addition, I suggest to divide the figures to better understand the results;
To perform a wb for: ERK1/2 and P-ERK1/2 for each times to add in figure 5;
To perform a wb for BMPRIa to add in figure 6;
To perform a wb for SMAD1/5 and P-SMAD1/5 to add in figure 7;
To improve the quality of IF in figure 9;
In order to underline the important of the data obtain, even if only of molecular basis, in the discussion section, I propose to add the hypothetical role that their evidences can have into improve the clinical approaches knowledge used today for therapy.
Round 2
Reviewer 2 Report
Dear Authors,
for your responses to my suggestions.
Finally, I suggest only to check the order of the new figures and the english language and style.
In my opinion the manuscript can be accepeted for the publication in JDB.